# A New Subform? Fast-Progressing, Severe Neurological Deterioration Caused by Spinal Epidural Lipomatosis

**DOI:** 10.3390/jcm11020366

**Published:** 2022-01-12

**Authors:** Thiemo Florin Dinger, Maija Susanna Eerikäinen, Anna Michel, Oliver Gembruch, Marvin Darkwah Oppong, Mehdi Chihi, Tobias Blau, Anne-Kathrin Uerschels, Daniela Pierscianek, Cornelius Deuschl, Ramazan Jabbarli, Ulrich Sure, Karsten Henning Wrede

**Affiliations:** 1Department of Neurosurgery and Spine Surgery, University Hospital of Essen, University of Duisburg-Essen, 47057 Duisburg, Germany; anna.michel@uk-essen.de (A.M.); oliver.gembruch@uk-essen.de (O.G.); marvin.darkwahoppong@uk-essen.de (M.D.O.); Mehdi.Chihi@uk-essen.de (M.C.); Ann-Kathrin.Uerschels@uk-essen.de (A.-K.U.); Daniela.Pierscianek@uk-essen.de (D.P.); Ramazan.Jabbarli@uk-essen.de (R.J.); Ulrich.Sure@uk-essen.de (U.S.); Karsten.Wrede@uk-essen.de (K.H.W.); 2Institute for Diagnostic and Interventional Radiology and Neuroradiology, University Hospital Essen, University of Duisburg-Essen, 47057 Duisburg, Germany; maija.eerikaeinen@uk-essen.de (M.S.E.); Cornelius.Deuschl@uk-essen.de (C.D.); 3Institute of Neuropathology, University Hospital of Essen, University of Duisburg-Essen, 47057 Duisburg, Germany; Tobias.Blau@uk-essen.de

**Keywords:** acute paraparesis, spinal cord injury, spinal epidural lipomatosis, spine, pathophysiology, neurological outcome, spinal surgery, atypical fat depositions

## Abstract

Spinal epidural lipomatosis (SEL) is a rare condition caused by hypertrophic growth of epidural fat. The prevalence of SEL in the Western world is approximately 1 in 40 patients and is likely to increase due to current medical and socio-economic developments. Rarely, SEL can lead to rapid severe neurological deterioration. The pathophysiology, optimal treatment, and outcome of these patients remain unclear. This study aims to widen current knowledge about this “SEL subform” and to improve its clinical management. A systematic literature review according to the PRISMA guidelines using PubMed, Scopus, Web of Science, and Cochrane Library was used to identify publications before 7 November 2021 reporting on acute/rapidly progressing, severe SEL. The final analysis comprised 12 patients with acute, severe SEL. The majority of the patients were male (9/12) and multimorbid (10/12). SEL mainly affected the thoracic part of the spinal cord (11/12), extending a median number of 7 spinal levels (range: 4–19). Surgery was the only chosen therapy (11/12), except for one critically ill patient. Regarding the outcome, half of the patients regained independence (6/11; = modified McCormick Scale ≤ II). Acute, severe SEL is a rare condition, mainly affecting multimorbid patients. The prognosis is poor in nearly 50% of the patients, even with maximum therapy. Further research is needed to stratify patients for conservative or surgical treatment.

## 1. Introduction

Spinal epidural lipomatosis (SEL) is defined by pathological hypertrophy of epidural benign fat cells leading to compression of neural structures [1]. As a consequence of ongoing medical and socio-economic development in the western world, the present prevalence of SEL (1 of 40 patients who underwent dedicated magnetic resonance imaging (MRI)) will increase along with multimorbidity and metabolic diseases [2]. The exact pathophysiology of SEL remains largely unknown. However, it is regarded as a consequence of a deranged fat metabolism (e.g., caused by endogenous or exogenous steroid excess), leading to atypical fat depositions [1]. Furthermore, venous congestion has been proposed as a leading contributing etiological factor for the severe and acute SEL subform with rapidly progressive symptoms [3,4].

The correlation of corticosteroid usage and SEL is significant because patients who require (chronic) corticosteroid treatment are often severely ill and represent a fragile subpopulation [5]. Choosing optimal therapy is difficult, as general treatment guidelines are pending. Physicians can only rely on case reports and reviews [1]. Furthermore, there is even less knowledge available on how to treat the acute SEL cases [6]. When faced with rapidly progressing neurological decline, physicians are thereby left with little choice but to offer maximum therapy (i.e., surgery) [7]. With the current study, we analyzed all published patients suffering from acute, severe SEL and additionally presented the experience gained from treating a patient at our facility. By improving the knowledge about the patients’ characteristics, symptoms, therapy, and outcome, we hope that future decisions on how to treat acute, severe SEL will be based on sounder evidence than currently possible.

## 2. Materials and Methods

### 2.1. Systematic Review, Search Strategy, and Acquisition of the SEL Cohort Data

The systematic review was performed according to the PRISMA guidelines [8]. PubMed, Scopus, Web of Science, and Cochrane Library databases were searched to retrieve all studies published before 7 November 2021 that reported on SEL patients with acute and rapidly progressing severe symptoms. Severe symptoms were defined as either functional (para-)plegia or fecal and urinary incontinence. The definition of ‘’acute onset with rapid progression’’ was a new episode that worsened to a clinical presentation of ‘’severe symptoms’’ in ≤ 72 h. We used different combinations of the following keywords to select all eligible studies: ‘’spinal epidural’’ (OR ‘’spinal’’), “lipoma*” and ‘’fast *’’ (OR ‘’rapid*‘’ OR ‘’quick*‘’). The exact search terms are shown in Appendix A. After the exclusion of duplicate records, TFD and AM independently screened the titles and abstracts (and, if necessary, the full text) to assess the eligibility of the studies. Reference lists of relevant publications were screened for additional studies. The publications’ language was restricted to English.

Studies were eligible for the review if they (1) reported at least one case with SEL, which fulfilled the defined criteria of (2) a ‘’severe presentation’’, (3) a ‘’rapid progression/acute onset’’ and had no other contributing (spinal) pathologies (like acute vertebral fractures) and (4) contained any data on demographic, clinical, radiographic, and anatomic characteristics or therapy and outcome of SEL patients (Figure 1).

Based on the selected studies, we extracted and summarized all available data from previously reported SEL patients with acute onset of severe symptoms. The patient cohort was additionally screened by TFD to exclude potential double-listed cases. The quality of the studies was addressed by OG and RJ using an adapted quality assessment score (QAS) (Appendix A), which has been described in detail previously [9]. All information was extracted from case reports reducing the points given for “minimizing selection bias “to a minimum of 2 points (e.g., prospective, consecutive studies would have received a maximum of 8 points).

### 2.2. Data Analysis

#### 2.2.1. Data Collection

The following parameters of the patients and SEL were recorded for further analyses: age at diagnosis, sex, medical history, history of SEL, neurological symptoms, modified Carlson Comorbidity Index [10] (mCCI), spinal levels of SEL, treatment, histopathological record, outcome (using the modified McCormick scale [11,12]), and time until severe symptoms, until diagnosis, and until treatment. Data extraction was performed by TFD and controlled by AM.

#### 2.2.2. Study Endpoints and Statistical Analysis

The defined primary endpoint was the assessment of associations between the recorded demographic, clinical, and anatomic characteristics of SEL patients and their outcomes. With only eleven identified individuals, the data analysis was restricted to descriptive analysis.

### 2.3. Image Analysis

According to Borré et al., anterior-posterior diameter ratios of epidural fat (EF) to the dural sack (DuS) and the spinal canal (SpiC) were calculated using Centricity™ Universal viewer (GE, Healthcare, North Richland Hills, TX, USA) [13]. Likewise, cross-section area ratios for EF vs. DuS and SpiC were calculated (see Equations (1)–(4) and Figure 2). For all ratio calculations, only axial planes of maximum compression level were used.
(1)$$\;\;\;Linear\;index_{DuS/EF}=\frac{D_{DuS}}{D_{EF}}$$
(2)$$\;\;\;Linear\;index_{EF/SpiC}=\frac{D_{EF}}{D_{SpiC}}$$
(3)$$\;\;\;Area\;index_{DuS/EF}=\frac{A_{DuS}}{A_{EF}}$$
(4)$$\;\;\;Area\;index_{EF/SpiC}=\frac{A_{EF}}{A_{SpiC}}$$
where *D* = anterior-posterior diameter, *A* = cross-sectional area, *DuS* = dural sack, *EF* = epidural fat, and *SpiC* = spinal canal.

## 3. Results

### Systematic Review of the Literature

Driven by the rapid progression and severity of symptoms, we felt the need to systematically identify all documented cases of acute, severe SEL to gather all accessible information to improve the management of this SEL subform.

A total of 11 patients who suffered from acute, severe SEL could be identified through the systematic review of the literature [6,7,14,15,16,17,18,19,20,21,22]. In addition, there was a prospectively recorded case from our clinic (Appendix A), giving a total of 12 patients for analysis (Table 1). Appendix A A lists the patients’ neurological symptoms at admission. Ten out of the twelve patients suffered from functional paraplegia, and nine suffered from at least incomplete cauda equina syndrome (bladder/rectal dysfunctions). Altogether, half of the patients fulfilled both criteria of a severe neurological disturbance (functional paraplegia and (incomplete) cauda equina syndrome).

Ten out of 12 patients were male, and the median age was 50.5 years (range: 20 to 69 years). Five patients had a medical history of either endogenous or exogenous steroid excess. Only three patients were obese (BMI > 30 kg/m^2^). The median mCCI score was two (range: 0–10), with ten patients meeting the criteria of multimorbidity, and three patients had a history of malignant neoplasm.

Regarding diagnostic management, all patients received laboratory workup. In five centers, cerebrospinal fluid analysis was performed to screen for neurological disease (Guillain-Barré syndrome, transverse myelitis, etc.). To investigate the etiology, all centers used imaging. Most of the centers used MRI (9/12), but in older studies, myelography was an alternative (3/12). In some centers, a CT was performed prior to (3/12) or instead of an MRI (post-myelography; 2/12) due to its better accessibility, especially in the past.

The number of involved spinal levels ranged from 4 to 19 (median of 7 levels), with no publication quantifying axial extension of SEL nor compression of the spinal cord (see Figure 2). All cases involved the thoracic spine, except for one solely lumbar SEL case, while the cervical and lumbar spine were involved in one case. Figure 3 depicts the extent of SEL in case ID#12. In this particular case, imaging allowed comparison of the extent of SEL in both the asymptomatic and symptomatic state (see Figure 2).

Except for one patient (ID#5), who died soon after the diagnosis was revealed by MRI [6], all other cases were treated surgically (see Figure 4A–C). Decompression was performed in all these cases, with additional removal of the SEL in all surgical cases (exemplarily Appendix A), except for one study that did not report on SEL removal. In one patient (ID#12), an additional dorsal fixation was performed to avoid instability caused by multiple-level laminectomy. Regarding the time from first symptoms to surgical intervention, there was a median of 60 h (range: 12–120 h). For four patients, no time until treatment was reported. Seven cases reported a histopathological confirmation of the diagnosis (see Figure 4D).

To analyze the outcome, the patients’ neurological status at admission versus follow- up/discharge (f/u) was quantified and compared using the mMcCS (Table 1). For one patient (ID#9), no outcome was reported [7]. Regarding the post-therapeutic recovery, two patients (ID#6 and ID#11) improved by one point and another two patients (ID#8 & ID#10) by two points in mMcCS. An improvement of three points and four points was observed in two further patients, respectively (ID#2 and ID#7). Four patients did not recover at all (ID#1, ID#3–5).

Additionally, the regained level of independence was analyzed (= mMcCS ≤ II(-III)) to estimate the “daily life” therapeutic benefit. Five patients did not regain independence, and four of these patients died during hospitalization or during a period of less than 2.5 months after treatment.

## 4. Discussion

These often multimorbid cases, with a rapidly progressing severe paraparesis due to SEL, highlight the main problems when managing these patients. In contrast to the sporadic clinical presentation of this subform, SEL typically shows a slow progression, rarely causing a high level of impairment. Generally, the therapeutic recommendations for SEL are based on case reports, case series, and reviews. Furthermore, no treatment recommendations are available for the management of acute, severe SEL.

Clinical diagnosis of SEL is difficult, being a rare disease. There are many more common causes for similar symptoms, including spinal disc herniation, inflammatory disease of the spinal cord (e.g., transverse myelitis, multiple sclerosis, Guillain-Barré syndrome), syringomyelia, and intraspinal tumors [23]. Therefore, the diagnosis can be delayed or incorrect [24,25,26]. Thankfully, SEL can be easily detected in both CT and MRI as a fat isodense (CT) or fat isointense (MRI) mass. By comparison, spinal angiolipoma is the most difficult of the multiple differential diagnoses to distinguish in unenhanced images.

Furthermore, angiolipomas show vivid contrast enhancement, whereas SEL does not enhance [27,28]. However, epidural fat tissue can easily be misinterpreted as an insignificant finding in the imaging, especially when the patient has no symptoms. In our patient, the pre-existing SEL showed significant growth over 17 months during the cancer therapy and corticosteroid exposure. Therefore, it can be hypothesized that earlier appreciation of the condition might have prevented the acute onset paraparesis and saved the patient from extensive surgery. Consequently, it is crucial that the treating physicians, as well as the radiologists, are aware of this diagnosis and the patient subgroup at risk.

Acute, fast-progressive SEL pathogenesis is barely understood, making it even more challenging to treat. Some authors reported a venous stasis in the epidural venous plexus with or without thrombosis as the main cause for acute worsening [3,4,22]. In contrast, others hold the compression by the epidural fat responsible [1]. Exact knowledge of the pathogenesis would have a significant impact on therapy, as the two above-mentioned causes require different treatments. A better understanding of the general pathophysiology would help differentiate which subgroup of patients might profit from a conservative therapy, even when dealing with acute and severe symptoms. The current literature suggests deranged (fat) metabolism, often associated with an increased endogenous or exogenous corticosteroid excess, as a major risk factor for SEL [1,23]. In particular, a neuro-sympatic auto-regulation of the fat metabolism is discussed, leading to an augmented fat deposition in non-physiological locations [29]. In addition, for two patients included in this review (ID#2 and 12), a systematic, augmented, atypical fat deposition was reported (see Figure 3). We observed an increase in the mediastinal and subcutaneous fat as well as fatty atrophy of the muscles. These observations support the hypothesis of a general disturbance of fat metabolism in this SEL subform.

Regarding the treatment, all published cases of acute, fast-progressive, and severe SEL without contraindication were treated surgically (representing maximum therapy), presumably due to severe clinical presentation and lack of knowledge about the effectiveness of non-surgical options (Table 1). Interestingly, two case reports demonstrated that conservative treatment could result in a good outcome, even in patients suffering from severe symptoms [29,30]. Lynch and colleagues reported on a case of severe paraparesis in a patient taking corticosteroids for a recently diagnosed ulcerative colitis, with partial recovery even after the first month after discontinuation of the treatment [30]. Bodelier et al. reported on an acute, rapidly progressive paraparesis and parahypesthesia sub-T5 due to severe SEL accompanied by a vertebral fracture in a patient suffering from an ectopic Cushing’s syndrome. Severe steroid-induced osteopenia was, in this case, a contraindication for back surgery. The patient underwent resection of the ACTH-producing tumor. In the 6-month follow-up after surgery, the patient showed a good recovery, with no difficulties in his routine life, and he was able to walk, ride a bike, and work again [29]. Both articles are of substantial importance, demonstrating that conservative management of patients with severe symptoms might be a good alternative in well-chosen cases.

This knowledge offers a new perspective, considering the mono-directional decision-making revealed by the systematic literature review (Table 1). The implication of surgery has to be considered. The patients often undergo multiple-level decompression (T3–T9), requiring additional dorsal fixation (see Appendix A). In addition, the systematic literature review revealed a median number of four involved spinal levels (Table 1), demonstrating a high risk for instability in the case of surgical treatment.

However, it has to be acknowledged that the combination of major surgery and often multimorbid SEL patients leads to high perioperative risk and potentially fatal complications [1]. Conclusively, in our systematic review, a median mCCI score of two (range: 0 to 10) was observed, representing a subpopulation of impaired state of health, which has been shown by others to correlate with a greater perioperative risk of complications and mortality than a normal mCCI [31,32]. Therefore, it is not surprising that we observed high mortality.

Nevertheless, it has to be mentioned that the information gathered here should be interpreted with caution. Systematic reviews of the literature are susceptible to selection, confounding, and information bias.

## 5. Conclusions

Overall, in our opinion, after the analysis of all cases, it can be stated that (I) acute, severe SEL is a potential new subform and is still insufficiently understood; (II) multimorbidity, resulting in a higher likelihood of complications, including death, seems to be overrepresented in these patients; (III) SEL should not be overlooked as a secondary finding in imaging of patients with risk factors; (IV) conservative treatment might be considered for well-chosen patients, as promising results have already been published.

## Figures and Tables

**Figure 1 jcm-11-00366-f001:**
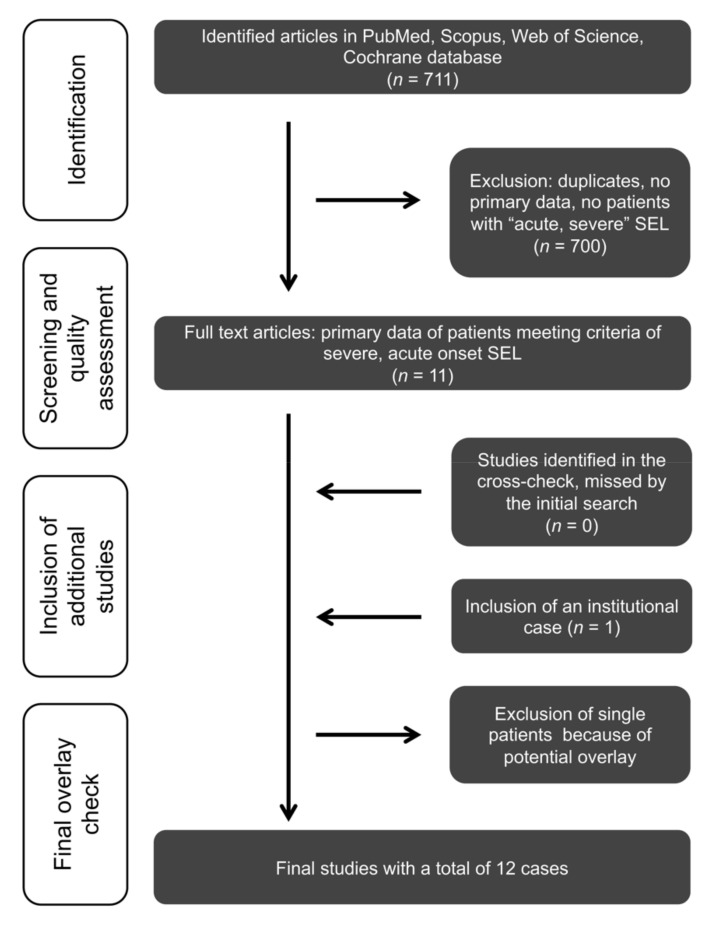
Flow chart of the systematic literature review. The inclusion and exclusion process of the initial 711 studies identified by PubMed, Scopus, Web of Science, and Cochrane database searches (Appendix A) are shown stepwise in the flow diagram. Abbreviation: SEL—spinal epidural lipomatosis.

**Figure 2 jcm-11-00366-f002:**
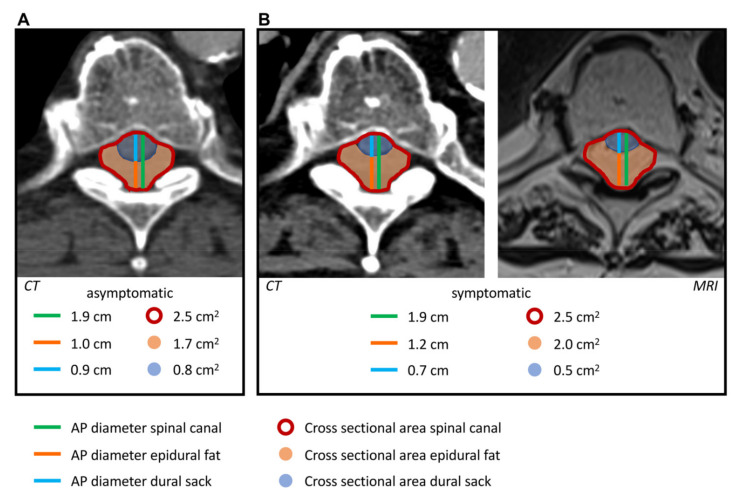
Exemplary illustration of quantification of SEL extent compared (1) to the nerval structures and (2) to the spinal canal. (**A**) The asymptomatic situation; (**B**) the symptomatic situation. Note that in the asymptomatic situation, the spinal canal is already almost 70% occupied by the SEL; an increase of 10% of the SEL thus results in functional paraplegia.

**Figure 3 jcm-11-00366-f003:**
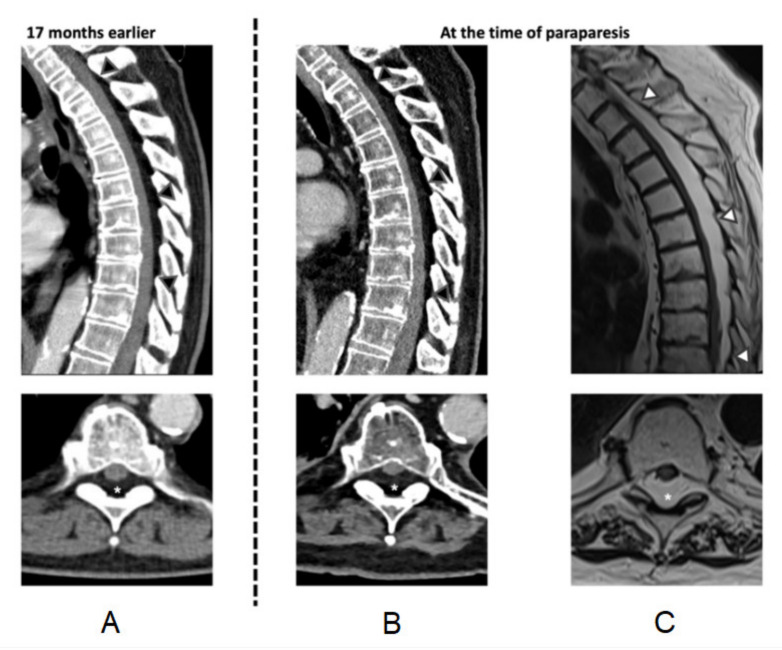
Preoperative imaging. (**A**) Retrospectively, SEL was already present in the CT performed during the diagnosis of lung cancer, 17 months earlier (sagittal and axial plane—at the level of T6). (**B**) Between the diagnosis of lung cancer and the paraparesis, the volume of the SEL increased significantly (preoperative CT in a sagittal and axial plane—at the level of T6) (see Figure 2). (**C**) Emergent MRI confirmed a fatty epidural tumor compressing the spinal cord (sagittal T1 TSE and axial T2 TSE at the level of T6), causing myelopathy at the level of T7–9 as shown by a low, patchy T2-hyperintense signal of the myelon at these levels. In addition, a general, atypical fat deposition pattern with an increase of epidural, mediastinal, and subcutaneous fat as well as fatty muscle atrophy was observed (**A** vs. **B**, **C**). The extent of SEL is highlighted in the sagittal planes by arrowheads and in the axial planes by asterisks. The mass severely compressed the spinal cord, which was ventrally displaced.

**Figure 4 jcm-11-00366-f004:**
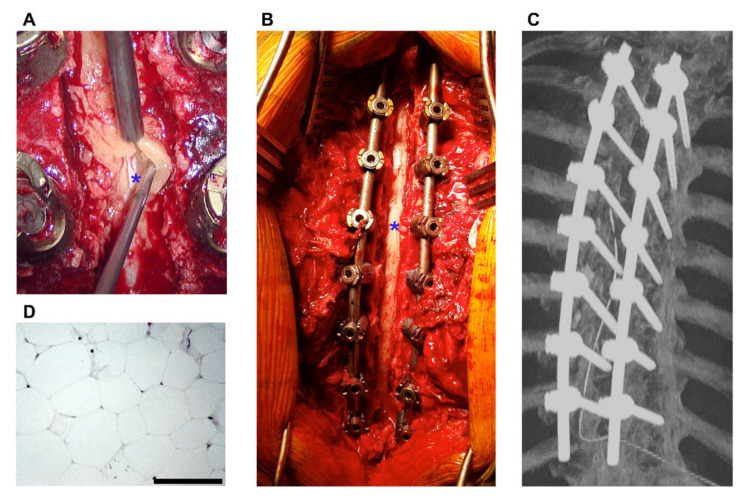
Intraoperative and postoperative images. (**A**) Intraoperative view of the microsurgical tumor removal. A section of SEL (yellowish mass) is lifted with an aspirator and a microsurgical punch from dura (marked with *). (**B**) Overview of the operation field at the end of tumor removal and spinal fixation, illustrating the extensions of the procedure. (**C**) Maximum intensity projection image of the postoperative CT scan. (**D**) H and E stained tumor slice after histopathological preparation, allowing the diagnosis of SEL with benign hypertrophic, unencapsulated fat cells (Bar represents 100 µm).

**Table 1 jcm-11-00366-t001:** Summary of patients with acute, severe SEL, with the first eleven patients representing the cases identified by the systematic literature review.

ID	Sex	Age	Medical History	Steroids	mCCI	mMcCS		Spinal Levels	Surgery	Histo	Complications	Time [h] till			Ref.	Year	QAS *
						ADM	f/u					Severe	Diagnosis	Treatment			
1	M	45	Hypothyroidism, obesity.	X	0	IV	IV	11	√	√	Death	48	48	N. r.	Toshniwal et al. [14]	1987	28
2	F	62	CAD, phlebitis, Raynaud’s syndrome, dermatomyositis.	√	0	V	I	5	√	X	No	“Rapidly progressive”	“Rapidly progressive”	N. r.	Buthiau et al. [15]	1988	28
3	M	52	Atopic dermatitis, CS with old fracture of T7.	√	0	V	-	4	√	X	Death	0	12	23	Kaplan et al. [16]	1989	28
4	M	20	None.	X	0	V	V	4	√	√	No	12	96	120	Meisheri et al. [17]	1996	28
5	M	27	NHL, BMT, GvHD, pneumonia, CS, obesity.	√	2	IV	-	19	X	X	Death	“few days”	“few days”	N.t.	Resnick et al. [6]	2004	24
6	M	41	HIV, metastasized NSCLC.	√	10	III	II	9	√	X	No	72	72	84	Vince et al. [18]	2005	30
7	M	60	CAD, Paget’s disease.	X	2	IV	I	6	√	√	No	72	72	72	Oikonomou et al. [19]	2007	30
8	M	55	AHT.	X	0	IV	II	5	√	√	No	24	48	48	López-González et al. [20]	2008	30
9	M	49	DM I, hepatitis C, i.v. heroin abuse, tobacco (30py).	X	2	N.r.	N.r.	13	√	X	No	12	12	12	Birmingham et al. [7]	2009	26
10	F	35	DM I, renal disease, endocarditis.	X	4	IV	II	9	√	√	No	“wake up”	12	12	Stephenson et al. [21]	2014	28
11	M	69	Obesity, COPD, DM II, AHT, hypercholesterolemia.	X	4	II	≤ II	2	√	√	No	“acute”	N.r.	N.r.	Tardivo et al. [22]	2021	26
12	M	67	Adiposity, DM II, alcohol abuse, tobacco (40py), NSCLC.	√	8	IV	III	8	√	√	Wound dehiscence, death	24	72	72	Appendix A	2022	32

Abbreviations: ADM–admission; AHT–arterial hypertension, BMT–bone marrow transplantation; CAD–coronary artery disease; CS–Cushing’s syndrome; DM–diabetes mellitus; f/u–follow up; GvHD–graft versus host disease; mCCI–modified Charlson comorbidity index; mMcCS–modified McCormick scale; NHL–Non-Hodgkin lymphoma; n.r.–not reported; NSCLC–non-small-cell lung carcinoma; n.t.–not treated; QAS–quality assessment score; Ref.–references; Appendix A. * See Appendix A.

## Data Availability

The data presented in this study are available on request from the corresponding author. The data are not publicly available because of sensitive patient information.

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
