# Peer review of "A New Subform? Fast-Progressing, Severe Neurological Deterioration Caused by Spinal Epidural Lipomatosis"

_jcm, 2022, doi:10.3390/jcm11020366_

Round 1
Reviewer 1 Report
The manuscript “A New Subform?–Fast-Progressing, Severe Neurological Deterioration Caused by Spinal Epidural Lipomatosis” is an article that aimed to widen current knowledge about this “SEL- subform” and to improve its clinical management.
Below are my comments and remarks regarding the article:
1. What was the time from the onset of symptoms to surgical intervention?
2. What was the preoperative procedure.
Reviewer 2 Report
Thank you to the authors for their interesting review of the literature on a challenging pathology to treat. The content seem OK but there are a lot of typos.
For fig 2 it would be helpful to tell us if there is significant statistical difference in the amount of epidural fat for asymptomatic versus symptomatic. Also can you describe the software used to calculate the volume so it may used at other centers.
Round 2
Reviewer 2 Report
Thank you for your corrections and contribution to the literature.
This manuscript is a resubmission of an earlier submission. The following is a list of the peer review reports and author responses from that submission.